# The Pharmacogenetics of Treatment with Quetiapine

**María Ortega-Ruiz** [1], **Paula Soria-Chacartegui** [1], **Gonzalo Villapalos-García** [1], **Francisco Abad-Santos** [1,2,*] and **Pablo Zubiaur** [1,2,*]

1    Clinical Pharmacology Department, Hospital Universitario de La Princesa, Instituto Teófilo Hernando, Instituto de Investigación Sanitaria La Princesa (IP), Universidad Autónoma de Madrid (UAM), 28006 Madrid, Spain; mariaortegaruiz2000@gmail.com (M.O.-R.); paula.soria@uam.es (P.S.-C.); g.villapalos@salud.madrid.org (G.V.-G.)

2    Centro de Investigación Biomédica en Red de Enfermedades Hepáticas y Digestivas (CIBERehd), Instituto de Salud Carlos III, 28006 Madrid, Spain

*    Correspondence: francisco.abad@uam.es (F.A.-S.); pablo.zubiaur@uam.es (P.Z.); Tel.: +34-915-202-425 (F.A.-S.); Fax: +34-915-202-540 (F.A.-S.)

**Abstract:** Quetiapine is a second-generation antipsychotic used for the treatment of schizophrenia, depression and bipolar disorder. The aim of this traditional review was to summarize the available pharmacogenetic information on this drug and to conclude about its clinical relevance. For this purpose, bibliographic research was performed in the Pharmacogenomics Knowledge Base (PharmGKB) database. A total of 23 articles were initially retrieved, of which 15 were finally included. A total of 19 associations were observed between 15 genes, such as *CYP3A4*, *CYP3A5*, *COMT* or *MC4R*, and 29 clinical events. No associations were consistently replicated between pharmacogenetic biomarkers and clinical events, except for that between the *CYP3A4* phenotype and quetiapine exposure, which was the only one considered clinically relevant. Consistently, the DPWG published a clinical guideline on this association, where dose adjustments for *CYP3A4* poor metabolizers (PMs) are indicated to prevent the occurrence of adverse drug reactions (ADRs).

**Keywords:** quetiapine; pharmacogenetics; second generation antipsychotic; PharmGKB

## 1. Introduction

Quetiapine is a second-generation antipsychotic (SGA) drug used for the treatment of schizophrenia and other mental disorders, such as depression or bipolar disorder (BD). For the latter indication, it is specifically used for the treatment of moderate to severe manic episodes, major depressive episodes and for the prevention of the recurrence of manic or depressive episodes in patients with type II BD who have previously responded to treatment with quetiapine [1]. It is also used for the treatment of generalized anxiety disorder (GAD) in monotherapy. Although some studies suggest that SGAs are not effective for GAD, quetiapine seems to relieve some of the symptoms [2]. Additionally, it seems to be useful in patients with obsessive–compulsive disorder treated with selective serotonin reuptake inhibitors [3]. It is administered in tablets, as a solution or suspension with different doses depending on patient demographics, disease severity, etc. The dosage is titrated at the beginning of treatment to avoid the risk for adverse drug reactions (ADRs). The initial dose is 50 mg, increasing over the following days to 300 mg, and then it is adjusted according to the clinical response and tolerability of the patients. The effective dose varies between 300–800 mg/day [1].

Administered orally, it is rapidly absorbed and extensively metabolized with an oral bioavailability of 9% due to first pass metabolism [1]. Food has no impact on drug absorption [4], and it shows linear pharmacokinetics in the approved dosage range (100 to 375 mg twice daily or 75 to 250 mg three times daily). Quetiapine is around 83% bound to plasma proteins [4]. It is extensively metabolized in the liver by the cytochrome P450 (CYP) system, presenting less than 5% of the original drug in the excretions, mostly in

urine (approximately 73%) and feces (21%) [4,5]. *CYP3A4* metabolizes 89% of quetiapine's dose, and the other 11% is metabolized mainly by *CYP2D6* and other enzymes, namely *CYP3A5*, *CYP2C9* and *CYP2C19* [1,6]. Eleven metabolites are known, some of which are active and others of which are inactive. They can be divided into three different groups: hydroxylated or sulfoxylated metabolites, dealkylated metabolites and a combination of the two reactions. The reaction between *CYP3A4* and quetiapine forms *N*-desalkyl quetiapine, also known as norquetiapine, which is further metabolized by *CYP2D6* to form 7-hydroxy *N*-alkylated quetiapine. *CYP2D6* directly transforms quetiapine into 7-hydroxy quetiapine (Figure 1) [6,7]. All of these are active metabolites. Quetiapine's half-life (t$_{1/2}$) is 7 h approximately, and its maximum concentration (C$_{max}$) is reached 1–2 h after oral administration (t$_{max}$). Furthermore, quetiapine is a *CYP1A2*, *CYP2C9*, *CYP2C19*, *CYP2D6* and *CYP3A4* inhibitor [1]. Additionally, recent studies suggest the participation of the catechol O-methyltransferase (*COMT*) enzyme in quetiapine's intermediate metabolism (Figure 1) [5].

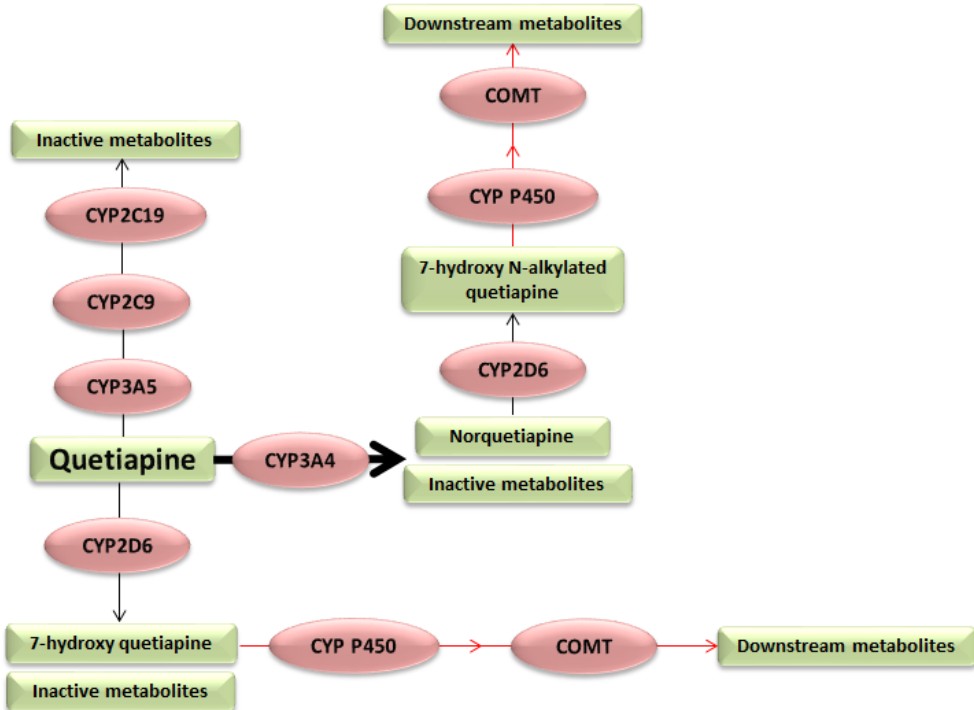

**Figure 1.** Genes involved in quetiapine metabolism. Red arrows indicate a recently suggested pathway that requires additional studies. *CYP3A5*: Cytochrome P450 Family 3 Subfamily A Member 5, *CYP2C19*: Cytochrome P450 Family 2 Subfamily C Member 19, *CYP2C9*: Cytochrome P450 Family 2 Subfamily C Member 9, *CYP3A5*: Cytochrome P450 Family 3 Subfamily A Member 4, *CYP3A5*: Cytochrome P450 Family 2 Subfamily D Member 6, *COMT*: Catechol-O-methyltransferase.

Although its precise mechanism of action remains controversial, it is a serotonin 5-HT2 receptor antagonist (HTR2) and a dopamine D1 and D2 receptor (DRD1 and DRD2) antagonist with affinity for other receptors, such as histamine H1, muscarinic M1, M3 and M5, α1-adrenergic and other serotonin receptors. On the other hand, norquetiapine has moderate to high affinity for several muscarinic receptors, which may explain the anticholinergic effects (Table 1) [1]. Quetiapine also inhibits the norepinephrine transporter (NET), mainly due to norquetiapine action [8]. The blockade of DRD2 in the mesocortical and mesolimbic pathways is proposed as the interaction responsible for the treatment of schizophrenia, where increased dopamine levels are responsible for negative and positive symptoms, respectively. 5-HT2 and α2 receptor antagonism is related to quetiapine's antidepressant activity, as well as noradrenaline transporter blockage by norquetiapine (Table 1) [1,7].

**Table 1.** Proposed pharmacodynamics explaining the mechanism of action of quetiapine and norquetiapine.

| Receptor | Norquetiapine Activity | Quetiapine Activity | Effect |
|---|---|---|---|
| Serotonin 5-HT$_{1A}$ receptor | Partial agonism (high affinity *) | Partial agonism | Antidepressant and anxiolytic effects. |
| Serotonin 5-HT$_{2A}$ receptor | Antagonism (high affinity *) $ | Antagonism $ | Treatment of negative and cognitive symptoms, antidepressant activity. |
| Serotonin 5-HT$_{2C}$ receptor | Antagonism (high affinity *) | Antagonism | Possibly related to weight gain. |
| Serotonin 5-HT$_7$ receptor | Antagonism (high affinity *) | Antagonism | Possibly related to antidepressant activity. |
| Dopamine D2 receptor | Antagonism (moderate affinity) | Antagonism (moderate affinity) | Treatment of negative and positive symptoms. |
| Histamine H1 receptor | Antagonism (high affinity *) | Antagonism | Possibly related to weight gain and sedative effects. |
| α1 receptor | Antagonism | Antagonism | Possibly related to orthostatic hypotension. |
| α2 receptor | Antagonism | Antagonism | Antidepressant activity. |
| Norepinephrine transporter (NET) | Antagonism | | Antidepressant activity. |
| Muscarinic receptors | Antagonism (high affinity) | | Anticolinergic effects, possibly related to adverse atropinic effects. |

* compared to quetiapine. $ higher than DRD affinity.

The most typical ADRs caused by quetiapine may be classified into neurological and metabolic ADRs. The main neurological ADR is somnolence, whereas extrapyramidal symptoms (dystonia, akathisia and tardive dyskinesia) occur with less frequency [1]. Metabolic ADRs include elevated serum triglyceride levels, elevated total cholesterol (predominantly LDL cholesterol), decreased HDL cholesterol, increased blood glucose to hyperglycemic levels, weight gain, decreased hemoglobin or, with lower frequency compared to other antipsychotics, hyperprolactinemia [9]. Other less frequent symptoms are leukopenia, increased appetite, nightmares, suicidal behavior and severe skin reactions. In pediatric populations, the most frequent ADRs are increased appetite, vomiting and an increase in blood pressure [1].

For several years now, different associations between genetic polymorphisms and the safety or effectiveness of antipsychotics have been published. To date, some of them have had enough replication, which justified the publication of pharmacogenetic guidelines oriented to individualize treatment. For instance, the relationship between the *CYP2D6* phenotype and ADR incidence is well reported for aripiprazole [10]. This led to the publication of the Dutch Pharmacogenetic Working Group guideline on aripiprazole, where a dose reduction of 67% of the standard dose is recommended for poor metabolizers (PM) [10]. In relation to quetiapine, a clinical guideline about *CYP3A4* and quetiapine has recently been published [11].

Taking all of this into account, genetic polymorphism may determine quetiapine pharmacokinetics and pharmacodynamics, which could determine drug safety and effectiveness. The aim of this review was to provide an assessment of the pharmacogenetics of quetiapine to compile all available information on quetiapine to address the feasibility of the clinical implementation of pharmacogenetic testing.

**2. Methods**

A systematic search was conducted in The Pharmacogenomics Knowledge Base (PharmGKB) of the different studies evaluating the relationship between genetic polymorphisms located in relevant pharmacogenes and quetiapine effectiveness, safety or other clinically

relevant events [12]. All the articles indexed in quetiapine's clinical annotation section of PharmGKB were included as well as any prescribing information. Associations found to be statistically significant in each study are noted and reflected in the Results Section.

A total of 29 clinical annotations based on 23 articles were retrieved from PharmGKB (Supplementary Tables S1 and S2). A total of 19 genes were identified with 19 variants in them. These variants were related to different clinical events or phenotypes (i.e., drug effectiveness and tolerability, weight gain and side effects) in patients with schizophrenia, autism spectrum, psychotic, depressive, bipolar or schizoaffective disorders. One prescribing information annotation was recorded. Figure 2 shows a PRISMA flow chart [13] of the studies that were identified, screened, assessed for eligibility and included in this review.

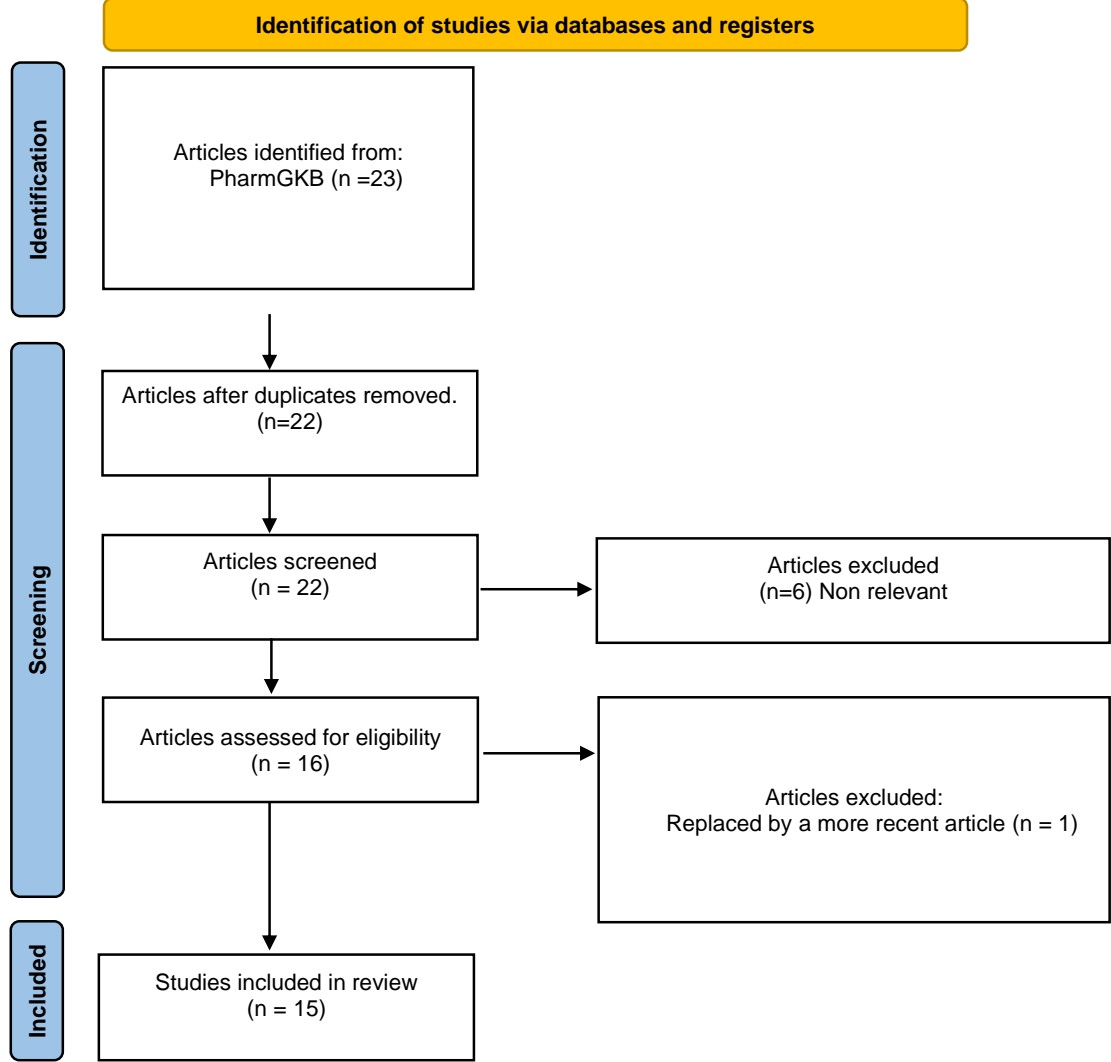

**Figure 2.** PRISMA flow chart of studies identified, screened, assessed for eligibility and included in this traditional review.

Among the 23 articles included, one was excluded, as it was replicated, and another 5 articles were excluded, as they did not directly refer to quetiapine. Another one was excluded, as it did not mention the specific gene variant studied. From the final 16 articles that were assessed for eligibility, one was excluded, as more recent research conducted by the same investigators was published on the same sample. The remaining articles (*n* = 15) were included in this review.

## 3. Results

Associations between genetic polymorphisms and the efficacy of quetiapine are shown in Table 2. Nine variants in five genes were identified. Moreover, associations related to the ADRs of quetiapine are shown in Table 3. Nine variants in eight genes were identified. Finally, associations between polymorphisms and quetiapine pharmacokinetic variability are shown in Table 4. Among all these studies, no replication was observed between any of the associations, neither among the same nor among different clinical events. In Table 2, it is important to highlight the high reliability in the three variants, according to the p-values obtained. The work of Cabaleiro et al. (2015) in Table 3 shows striking data, as they found significant associations in different cytochrome P450 genes. However, of all the results obtained, the most relevant data is *CYP3A4* shown in Table 4, as it already has a clinical guideline.

**Table 2.** Association between gene variants and their effect on drug efficacy [12].

| Gene | Variant | Population | Association | *p* Value | References |
|------|---------|-----------|-------------|-----------|------------|
| *COMT* | rs4818 G>C | 995 Patients with schizophrenia. | C allele was associated with a worse response (PANSS score) compared to G allele carriers. | 0.007 | Xu et al. (2016) [14] |
| | rs6269 G>A | | A allele was associated with a worse response (PANSS score) compared to G allele. | 0.008 | |
| | rs5993883 G>T | | T allele was associated with a worse response (PANSS score) compared to G allele. | 0.007 | |
| *EPM2A* | rs1415744 T>C | 573 Patients with schizophrenia. | C allele was associated with an increased response (PANSS score) compared to T allele. | 0.004 | Porcelli et al. (2016) [15] |
| *HTR1A* | rs10042486 T>C | 221 Patients with schizophrenia and 171 healthy controls. | CT and CC were associated with a worse response (PANSS score) compared to TT allele diplotype. | $1.0 \times 10^{-6}$ | Crisafulli et al. (2012) [16] |
| *PDE4D* | rs17742120 A>G | 738 Patients with schizophrenia. | G allele was associated with a decreased response (PANSS score) compared to A allele. | $1.58 \times 10^{-7}$ | Clark et al. (2013) [17] |
| | rs2164660 G>A | | A allele was associated with an increased response (PANSS score) compared to G allele. | $1.87 \times 10^{-7}$ | |
| | rs17382202 C>T | | T allele was associated with an increased response (PANSS score) compared to C allele. | $4.21 \times 10^{-8}$ | |
| *RGS4* | rs951439 C>T | 678 Patients with schizophrenia. | CC diplotype was associated with an increased response (PANSS score) compared to CT and TT allele. | 0.010 | Campbell et al. (2008) [18] |

\* PANSS: Positive and Negative Symptoms Scale. *COMT*: Catechol-O-methyltransferase, EPM2A: Epilepsy Progressive Myoclonus type 2A, HTR1A: 5-Hydroxytryptamine Receptor 1A, *PDE4D*: phosphodiesterase 4D, RGS4: Regulator of G Protein Signaling 4. Data retrieved from the Clinical Annotations section in PharmGKB.

**Table 3.** Association between gene variants and the adverse reactions of quetiapine [12].

| Gene | Variant | Population | Association | *p* Value | References |
|---|---|---|---|---|---|
| *FAAH* | rs324420 C>A | 83 Patients with psychotic episodes. | A allele was associated with increased weight gain compared to C allele carriers. | 0.002 | Monteleone, P. et al. (2010) [19] |
| *GRIN2B* | rs1806201 G>A | 211 healthy volunteers treated with a single dose of risperidone olanzapine or quetiapine. | AA diplotype was associated with an increased likelihood of adverse neuronal reactions compared to AG and GG diplotypes. | 0.025 | López-Rodríguez, R et al. (2013) [20] |
| *MC4R* | rs17782313 T>C | 345 patients with psychiatric episodes. | CC diplotype was associated with an increased likelihood of weight gain compared to CT and TT diplotypes. | 0.005 | Czerwensky, F et al. (2013) [21] |
| | rs489693 A>C | 139 patients with psychiatric episodes. | AA diplotype was associated with a decreased likelihood of ADRs compared to AC and CC diplotypes. | <0.01 | Malhotra, A. K (2012) [22] |
| | | 345 patients with psychiatric episodes. | | 0.017 | Czerwensky, F et al. (2013) [21] |
| *SH2B1* | rs388819 A>C | 357 patients with psychiatric episodes. | CC diplotype was associated with increased LDL levels compared to AA and AC diplotypes. | 0.005 | Delacrétaz, A et al. (2017) [23] |
| *RABEP1* | rs1000940 A>G | 357 patients with psychiatric episodes. | AG and GG diplotypes were associated with lower glucose concentrations compared to AA diplotype. | <0.001 | |
| *CYP2C19* | *2, and *4 | 79 healthy volunteers receiving a single dose of each quetiapine formulation. | *1/*2, *2/*2 and *2/*4 diplotypes were associated with higher prolactin plasma concentrations compared to *1/*1 diplotype. | 0.012 | Cabaleiro et al. (2015) [24] |
| *CYP2C9* | *2 and *3 | | *1/*2, *1/*3 and *2/*3 diplotypes showed somnolence. | 0.015 | |
| *CYP1A1* | *2 | | *1/*2 diplotype showed somnolence. | 0.020 | |
| *CYP1A1* | *2 | | *1/*2 diplotype showed neurological events. | 0.024 | |

*FAAH*: Fatty Acid Amide Hydrolase, *GRIN2B*: Glutamate Ionotropic Receptor NMDA Type Subunit 2B, *MC4R*: Melanocortin 4 Receptor, *SH2B1*: SH2B Adaptor Protein 1, *RABEP1*: Rabaptin, RAB GTPase Binding Effector Protein 1, *CYP2C19*: Cytochrome P450 Family 2 Subfamily C Member 19, *CYP2C9*: Cytochrome P450 Family 2 Subfamily C Member 9, *CYP1A1*: Cytochrome P450 Family 1 Subfamily A Member 1. The data have been adapted from that shown on the PharmGKB website.

**Table 4.** Association between genetic variants and the pharmacokinetic effects of quetiapine [12].

| Gene | Variant | Population | Association | *p* Value | References |
|---|---|---|---|---|---|
| *DRD3* | rs6280 C>T | 79 healthy volunteers receiving a single dose of each quetiapine formulation. | TT diplotype was associated with increased clearance compared to CC and CT diplotypes. | 0.030 | Cabaleiro et al. (2015) [24] |
| *CYP1A2* | *1,*1C | | *1C/*1C diplotype was associated with higher exposure to quetiapine compared to *1/*1 diplotype. | 0.067 | |

**Table 4.** *Cont.*

| Gene | Variant | Population | Association | *p* Value | References |
|------|---------|-----------|-------------|-----------|------------|
| *COMT* | rs13306278 C>T | 49 healthy volunteers treated with two doses of quetiapine. | T allele was associated with higher exposure to quetiapine compared to C allele. | 0.008 | Zubiaur et al. (2021) [5] |
| *CYP2B6* | PM | | PM showed higher $t_{1/2}$ compared to RM, NM or IM. | 0.005 | |
| *ABCG2* | rs2231142 G>T | | T allele was associated with quetiapine accumulation compared to G allele. | 0.027 | |
| *CYP3A5* | *3 | | *3/*3 showed higher $t_{1/2}$ compared to *1/*1 or *1/*3. | 0.018 | |
| *CYP3A4* | *22 | 238 patients treated with quetiapine. | *22/*22 or *22/*1 diplotypes were associated with increased exposure to quetiapine compared to *1/*1 diplotype. | 0.007 | Weide Karen et al. (2014) [25] |
| *CYP3A4* | *3,*20,*22 | 19 patients treated with a single dose of quetiapine. | *3,*20,*22 alleles were associated with increased exposure to quetiapine compared to *1 allele. | 0.099 | Saiz Rodriguez et al. (2020) [26] |
| *CYP3A5* | *3 | 40 healthy volunteers treated with a single dose of quetiapine. | *3/*3 diplotype was associated with increased exposure to quetiapine compared to *1/*3 or *1/*1 diplotypes. | 0.0017 | Kim, K.-A. et al. (2014) [27] |

The authors performed a haplotype analysis, which analyzed *CYP3A4*\*3 (rs4986910), *CYP3A4*\*20 (rs67666821) and *CYP3A4*\*22 (rs35599367). DRD3: Dopamine Receptor D3, *CYP1A2*: Cytochrome P450 Family 1 Subfamily A Member 2, *COMT*: Catechol-O-Methyltransferase, CYP2B6: Cytochrome P450 Family 2 Subfamily B Member 6, ABCG2: ATP Binding Cassette Subfamily G Member 2, *CYP3A4*: Cytochrome P450 Family 3 Subfamily A Member 4, *CYP3A5*: Cytochrome P450 Family 3 Subfamily A Member 5. The data has been adapted from that shown on the PharmGKB website.

Finally, the DPWG's guideline on *CYP3A4* and quetiapine was noted. Briefly, *CYP3A4* IM shows a lower conversion of quetiapine to active and inactive metabolites, with a 20% increase in quetiapine plasma concentrations, whose clinical effect does not require a dose adjustment [28]. Regarding *CYP3A4* PM, they are reported to show a 3.2-fold higher quetiapine plasma concentration. Therefore, a dose reduction to 30% or, if possible, changing the drug to a less *CYP3A4*-dependent drug is recommended, depending on the indication. This guideline [11] is based on the works of Weide Karen et al. and Saiz Rodriguez et al. [24,25].

## 4. Discussion

The implementation of drug-specific pharmacogenetic testing in the management of psychiatric patients has been increasing rapidly in recent times [28]. Actionable pharmacogenetic biomarkers have been available for years for antipsychotics such as aripiprazole, risperidone, haloperidol, pimozide or zuclopenthixol. For instance, the implementation of *CYP2D6* testing before aripiprazole prescription leads to better management of adverse reactions [28]. All the pharmacogenetic guidelines published to date were issued by the DPWG, which recently published its recommendation on the phenotype of *CYP3A4* PM–quetiapine gene–drug interaction (Table 5). *CYP3A4* is, therefore, the only clinically relevant biomarker that can help individualize quetiapine pharmacotherapy based on the publication of pharmacogenetic guidelines. Furthermore, the Clinical Pharmacogenetics Implementation Consortium (CPIC) rated the drafting of a pharmacogenetic guideline on *CYP2D6* and antipsychotics (i.e., aripiprazole, brexpiprazole, pimozide, risperidone, iloperi-

done, perphenazine, clozapine, haloperidol, olanzapine, thioridazine, zuclopenthixol) with a B level of priority [29]. Apparently, this guideline will focus on the association of these drugs with *CYP2D6* polymorphism. This may be the reason why quetiapine is not included, as the interaction between *CYP2D6* and quetiapine exposure was previously considered irrelevant [30]. It would be expected that the *CYP3A4*–quetiapine interaction was included in the new CPIC guidelines.

**Table 5.** Pharmacogenetic guidelines published to date on antipsychotic pharmacotherapy.

| Drug | Phenotype | Implications | Recommendations | Authority |
|---|---|---|---|---|
| Quetiapine (2022) [31] | *CYP3A4* PM | Reduced *CYP3A4* activity. | Use 30% of the standard dose or choose another alternative that is not metabolized by *CYP3A4*. | DPWG |
| Risperidone (2020) [32] | *CYP2D6* UM | Higher ratio of the active metabolite. | Choose another antipsychotic or titrate the dose according to the maximum dose for the active metabolite. | DPWG |
| | *CYP2D6* PM | Increased risperidone plasma concentration. | Reduce to 67% of the standard dose. | DPWG |
| Aripiprazol (2021) [33] | *CYP2D6* PM | Increased risk of ADRs. | Administer no more than 10 mg/day or 300 mg/month. | DPWG |
| Haloperidol (2021) [34] | *CYP2D6* UM | Increased conversion of haloperidol. | Administer 1.5 times the standard dose or choose an alternative drug. | DPWG |
| | *CYP2D6* PM | Decreased conversion of haloperidol. | Administer 60% of the standard dose. | DPWG |
| Zuclopenthixol (2022) [35] | *CYP2D6* IM | Decreased conversion of zuclopentixol. | Administer 75% of the standard dose. | DPWG |
| | *CYP2D6* PM | Decreased conversion of zuclopentixol. | Administer 50% of the standard dose. | DPWG |
| Pimozide (2021) [36] | *CYP2D6* IM | Increased plasma concentration of pimozide. | Administer no more than 80% of the standard maximum dose. | DPWG |
| | *CYP2D6* PM | Increased plasma concentration of pimozide. | Administer no more than 50% of the standard maximum dose. | DPWG |

All recommendations are included in the Pharmacogenetics recommendations document of the Royal Dutch Society for the Advancement of Pharmacy (KNMP) Dutch Pharmacogenetics Working Group (DPWG).

Concerning *CYP3A5*, some level of association was observed (i.e., two independent studies observed the accumulation of the drug or higher $t_{1/2}$ in PM). This is consistent with the impact of the *CYP3A5* phenotype on the disposition of well-known substrates (e.g., tacrolimus [37]). Therefore, although additional studies are warranted to determine the clinical relevance of this gene–drug interaction, *CYP3A5* can be considered a good candidate biomarker to individualize quetiapine pharmacotherapy.

Regarding the remaining associations observed in the current work, no replication was observed among them; therefore, they cannot be concluded clinically relevant. According to the authors' preferences, some of them are described more in depth as follows.

Concerning quetiapine effectiveness, heterogeneous associations were observed for polymorphisms located in enzymes such as *COMT* but also for receptors apparently not involved in the quetiapine mechanism of action, such as *MC4R*. For instance, *COMT* is known to play an important role in the effectiveness of many antipsychotics due to its role in dopamine degradation, as the dopaminergic system is important in antipsychotic activ-

ity [14]. However, a recent study suggests that a portion of quetiapine may be metabolized by this enzyme [5]; in the latter work, the *COMT* rs13306278 T led to the accumulation of a metabolite (7,8-dihydroxy-*N*-desalkyl quetiapine), which was suggested to inhibit *CYP3A4*, causing the accumulation of quetiapine and therefore an increased risk for ADRs; nonetheless, additional studies are warranted to confirm this hypothesis [5].

With respect to associations with a response to quetiapine, *PDE4D* is worth mentioning, which is a cyclic AMP-specific phosphodiesterase expressed in the human brain. It was considered a potential candidate to determine drug effectiveness. Its inhibitors improve dopamine D1 receptor signaling and have an antipsychotic function [17]. *PDE4D* polymorphism was related to drug response variability, and the effect of different variants changed towards a better or worse response. However, additional studies aimed at replicating these associations are required. In relation to ADR occurrence, the most relevant gene identified in this section may be the one encoding for melanocortin receptor 4 (*MC4R*). It is a G-protein-coupled receptor involved in the hypothalamic leptin-melanocortin signaling pathway. It is involved in energy homeostasis but also in satiety regulation mechanisms [38]. Recent studies suggest that certain *MC4R* polymorphisms can cause a total or partial loss of its function [21]. In addition, it has been associated with SGA-related weight gain [21]. Likewise, additional research is warranted to clarify these associations.

**5. Conclusions**

To date, numerous studies have been published on quetiapine pharmacogenetics, with *CYP3A4* being the main pharmacogenetic biomarker. Therapy should be optimized based on the *CYP3A4* phenotype, which can reduce the incidence of ADRs and can increase drug adherence and, therefore, effectiveness. The *CYP3A5* phenotype conditions the exposure to quetiapine and may be related to ADR incidence. Additional studies are required to clearly determine the clinical relevance of the latter drug–gene interaction. Eventually, pharmacogenetic guidelines could also consider dose adjustments based on the *CYP3A5* phenotype. The remaining associations were considered not clinically relevant due to the low level of replication.

**Supplementary Materials:** The following supporting information can be downloaded at: https://www.mdpi.com/article/10.3390/futurepharmacol2030018/s1, Table S1: List of screened publications, ordered by year of publication.; Table S2: List of screened annotations ordered by gene.

**Author Contributions:** M.O.-R. and P.Z.: writing—original draft preparation, P.S.-C., G.V.-G. and F.A.-S.: writing—review and editing. All authors have read and agreed to the published version of the manuscript.

**Funding:** P.S.-C. was financed by the "Contrato Predoctoral para la Formación de Personal Investigador UAM 2021". G.V.-G. was supported by a PFIS predoctoral grant (FI20/00090), and P.Z. was financed by the "Contrato Margarita Salas de la convocatoria para la Recualificación del Sistema Universitario Español" (UAM).

**Institutional Review Board Statement:** Not applicable.

**Informed Consent Statement:** Not applicable.

**Data Availability Statement:** All data are available at PharmGKB (https://www.pharmgkb.org/ (accessed on 20 June 2022)).

**Conflicts of Interest:** F.A.-S. has been consultant or investigator in clinical trials sponsored by the following pharmaceutical companies: Abbott, Alter, Chemo, Cinfa, FAES, Farmalíder, Ferrer, GlaxoSmithKline, Galenicum, Gilead, Italfarmaco, Janssen-Cilag, Kern, Normon, Novartis, Servier, Silver Pharma, Teva and Zambon. The remaining authors declare no conflicts of interest.

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
