# Peer review of "The Pharmacogenetics of Treatment with Quetiapine"

_futurepharmacol, doi:10.3390/futurepharmacol2030018_

Round 1

Reviewer 1 Report

The authors have essentially summarised the data tables on PHARMGKB website. This aspect should be made clear and PHARMGKB acknowledged appropriately. 

Title:

Suggest – Pharmacogenetics of quetiapine treatment.

Abstract:

The abstract has multiple tenses (past/present/future) – and this should be reviewed.

Introduction:

Introduction is well written.

It would be beneficial to include a figure showing the pathways of quetiapine metabolism and the various genes involved.

As above the language and grammar needs review.

Methods:

This section needs more details.

An appendix table listing the 29 annotations/23 publications would be useful. Preferably ordered by year published. Then maybe italicize the ones excluded and bold the ones included-15

Inclusion criteria is described – what about exclusions?

Also, PharmGKB is not referenced in the methods. Please fix this.

Figure 1- should preferably appear in methods, rather than Results.

PRSIMA should be referenced.

Results:

I am unsure about the way the tables have been reproduced. They are very similar to the way they appear in PHARMGkb. Please reference PHARMGKB in each table and say the data has been adapted/modified from that shown on the website.

Discussion:

Table 4 – a further column indicating which authority published the guideline would be good – CPIC, DPWG etc.

The grammar and use of tenses throughout the manuscript is incorrect and requires review.

Author Response

Please find attached our responses.

Reviewer 2 Report

In my opinion, the article is valuable, well written and makes a significant contribution to clinical and scientific knowledge. I only have comments regarding the Introduction:

 1. The authors should mention not only the use of quetiapine in the treatment of schizophrenia, depression or bipolar disorder, but also the evidence of its effectiveness in GAD monotherapy, or the potentiation of the effect of antidepressants in obsessive-compulsive disorder.

 2. The authors write: “Although its precise mechanism of action remains controversial, it is a serotonin 5-HT2 receptor antagonist (HTR2) and a dopamine D1 and D2 receptor (DRD1 and DRD2) antagonist with affinity for other receptors such as histamine H1, muscarinic M1, M3 and M5, α1-adrenergic and other serotonin receptors.” I believe they should mention in more detail the various pharmacodynamic properties of quetiapine - including its activity on various serotonin receptors in more detail. It's best to put it in a table.

 3. The authors write: "Quetiapine also inhibits the norepinephrine transporter (NET)." However, it should be added that this mechanism is more significant, first of all, in the case of norquetiapine.

 4. The authors write: "5-HT2 antagonism is related to quetiapine's antidepressant activity". In this case, it is necessary to add that the antidepressant activity of quetiapine is also the effect of the afla2 receptor blockade and - to a significant extent - the effect of noradrenaline transporter blockade by norquetiapine.

 5. The description of the adverse reaction profile for quetiapine can be misleading. In the case of extrapyramidal or hyperprolactinemic symptoms, it must be mentioned that the risk of the above-mentioned side effects is, under normal conditions, extremely low for quetiapine and one of the lowest compared to most other antipsychotics.

Author Response

Please find attached our responses.
